   

# Identification of *Plasmodium* GAPDH epitopes for generation of antibodies that inhibit malaria infection

Sung-Jae Cha[1], Kyle Jarrod McLean[2], Marcelo Jacobs-Lorena[1]

***Plasmodium* sporozoite liver infection is an essential step for parasite development in its mammalian host. Previously, we used a phage display library to identify mimotope peptides that bind to Kupffer cells and competitively inhibit sporozoite–Kupffer cell interaction. These peptides led to the identification of a Kupffer cell receptor—CD68—and a *Plasmodium* sporozoite ligand—GAPDH—that are required for sporozoite traversal of Kupffer cells and subsequent infection of hepatocytes. Here, we report that the C-terminal end of *Plasmodium* GAPDH interacts with the Kupffer CD68 receptor, and identify two epitopes within this region as candidate antigens for the development of antibodies that inhibit *Plasmodium* infection.**

## Introduction

Malaria infection of the vertebrate host begins with the deposition of sporozoites in the skin by an infected mosquito. Sporozoites then enter the circulation and are quickly captured in the liver sinusoids via strong interactions between circumsporozoite protein (CSP) on the sporozoite surface and glycosaminoglycans (GAGs) on the liver sinusoid. The initial binding of the sporozoite CSP to the liver GAGs was first reported on the basolateral side of the hepatocytes (Frevert et al, 1993; Cerami et al, 1994). Follow-up studies determined that sporozoite CSP binds to GAGs synthesized by Stellate cells and that protrude into the vascular lumen through endothelial fenestrations (Pradel et al, 2002). This interaction is not obligatory for host cell recognition (Frevert et al, 1996). Microscopic studies indicated that sporozoites leave the circulation mainly by traversing Kupffer cells (Meis et al, 1983, 1985; Vreden, 1994; Pradel & Frevert et al, 2001), which are part of the sinusoid lining, followed by infection of underlying hepatocytes (Frevert et al, 2005). A later study using intravital confocal microscopy and a transgenic mouse line that expresses GFP in the endothelial cells, provided evidence that sporozoites can also leave the circulation by traversing endothelial cells (Tavares et al, 2013). More recently, Kupffer cell CD68 was identified as a sporozoite receptor, suggesting the molecular basis for the preferential sporozoite interaction with Kupffer cells. In CD68-knockout mice, invasion of the liver was reduced by more than 70% of sporozoites compared with wild-type mice (Cha et al, 2015).

Identification of CD68 as a sporozoite receptor stemmed from the selection of three peptides—P39, P61, and P52—from a screen of a phage library for peptides that bind to Kupffer cells (Cha et al, 2015). Not only do these peptides selectively bind to the Kupffer cell surface, but importantly, they also competitively inhibit sporozoite Kupffer cell entry and access to the liver parenchyma. Further experiments showed that these peptides bind to the CD68 protein on the surface of Kupffer cells and that they are mimotopes (structural mimics) of *Plasmodium* GAPDH (pGAPDH) present on the surface of sporozoites. CD68 and pGAPDH directly interact, and this interaction is crucial for sporozoite traversal of Kupffer cells (Cha et al, 2016).

In addition to its major role in glycolysis, GAPDH has been recognized as a vaccine candidate, as it occurs on the surface of several pathogenic microbes, where it serves as a ligand for host cell recognition and invasion (Pancholi & Fischetti et al, 1992, 1997; Argiro et al, 2000; Rosinha et al, 2002; Bergmann et al, 2004; Boël et al, 2005). Because GAPDH is conserved in evolution and its amino acid sequence is highly similar between pathogens and higher organisms, use of the full protein as a vaccine antigen is not practical. Thus, identification of parasite GAPDH-specific epitopes that can elicit protective antibody generation is a priority (Perez-Casal & Potter, 2016). Our study showed that immunization of mice with KLH-conjugated P39 peptide (a GAPDH mimotope) elicits strong protective immunity and that anti-P39 sera recognize the pGAPDH protein. Here, we report on epitope mapping assays using an anti-P39 antibody to identify domains of *Plasmodium berghei* GAPDH (PbGAPDH) that can act as protective epitopes. We show that protective epitopes unique to PbGAPDH are located within the C-terminus of the protein, in a region that is predicted to be in an exposed pocket in the folded protein.

[1]Department of Molecular Microbiology and Immunology, Malaria Research Institute, Johns Hopkins Bloomberg School of Public Health, Baltimore, MD, USA    [2]Department of Biological Engineering, Massachusetts Institute of Technology, Cambridge, MA, USA

Correspondence: mlorena@jhsph.edu

# Results

## Assessment of mimotope peptides P39, P61, and P52 as immunogenic antigens for protection from *P berghei* sporozoite infection

Using a phage display peptide library, we have previously identified three peptides—P39, P61, and P52—that bind to Kupffer cells, and by doing so, strongly inhibit sporozoite binding and entry (Cha et al, 2015). We also showed that immunization with the P39 peptide strongly protects mice from *P berghei* sporozoite infection by mosquito biting. Fig 1A confirms that the anti-P39 antibody recognizes a sporozoite protein with mobility identical to that of GAPDH. However, the other two peptides—P61 and P52—had not been characterized. We immunized mice with KLH-conjugated synthetic mimotope peptides. Antibodies to all three peptides recognize the recombinant pGAPDH (Fig 1B). However, each peptide appears to mimic a different PbGAPDH epitope (Fig 1C). Next, we assessed immunogenicity and protective potential of each candidate mimotope antigen. Mice were immunized with single or with different combinations of the three mimotope antigens. Whereas the total antibody concentration in each group did not differ significantly, specific immunogenicity of P39 was significantly higher than that of P52 (Fig 2A). Each immunization was followed by challenge with *P berghei* sporozoites via biting by two infected *Anopheles stephensi* mosquitos. Immunization with the P39 peptide had the strongest protective efficacy with 67% inhibition of sporozoite infection compared with KLH-immunized control groups (Fig 2B). These initial data suggest that P39 has the strongest protective potential, and therefore, it was used for all following experiments.

## An anti-P39 antibody recognizes the C-terminal end of PbGAPDH

To identify PbGAPDH protective epitopes, recombinant PbGAPDH protein fragments were produced as illustrated in Fig 3A. PbGAPDH has 337 amino acids encoding two conserved domains: [Gp_dh_N] 4-152, NAD-binding domain, and [Gp_dh_C] 157-317, C-terminal domain (Kim et al, 1995). Each of six ~65 amino acid subdomains were cloned and the corresponding proteins were tested for recognition by the anti-P39 antibody or by a polyclonal antibody produced by immunization with the full length PbGAPDH recombinant protein after enzymatic removal of the tag. The polyclonal anti-PbGAPDH antibody recognizes all fragments except for G1 and the pET tag, whereas the anti-P39 antibody recognizes only the G6 fragment. The pET tag, which is a component of each protein fragment, served as a loading control (Fig 3B). By contrast, the anti-P61 and anti-P52 antibodies mainly bound to the G3 fragment. To confirm specific recognition of the G6 fragment by the anti-P39 antibody, we incubated *P berghei* sporozoites with the anti-P39 antibody plus each PbGAPDH fragment. We hypothesized that the PbGAPDH fragment containing the anti-P39 recognition epitope would competitively inhibit antibody binding to GAPDH on the surface of the sporozoite. As shown in Fig 3C, the G6 fragment specifically inhibited anti-P39 antibody binding to the sporozoite surface. These results were confirmed

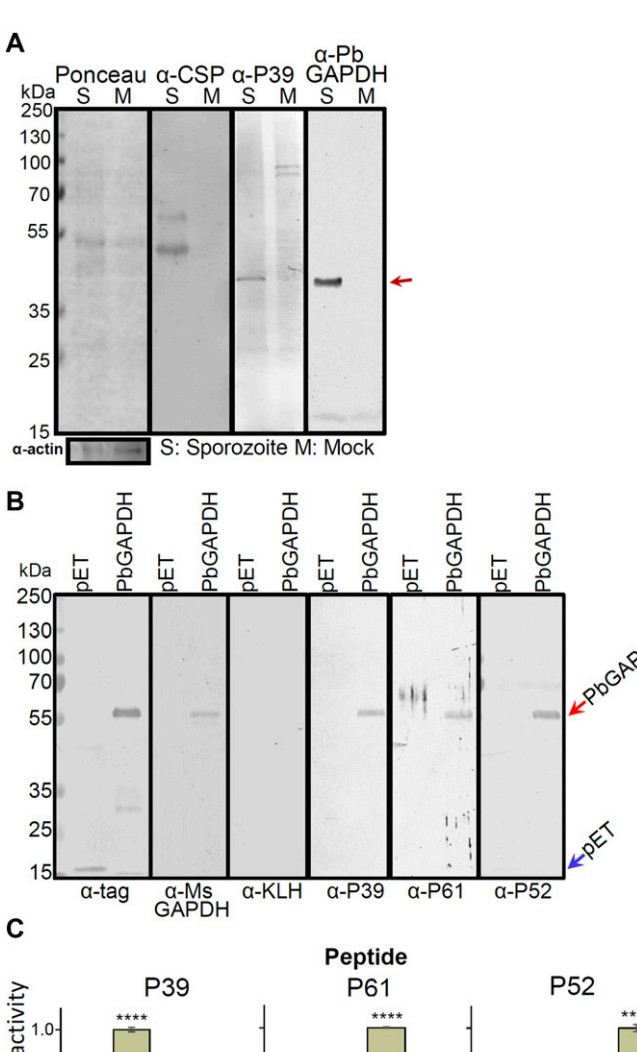

**Figure 1. Three different mimotope peptides mimic different domains of the the *P* berghei GAPDH molecule.**
**(A)** The anti-P39 antibody specifically recognizes PbGAPDH in Western blots of *P berghei* sporozoite lysates. Ponceau stain and the anti-actin antibody were used as loading controls. "S": lysates of sporozoites purified from infected *A stephensi* salivary glands; "M": mock lysates obtained from uninfected *A stephensi* salivary glands. Anti-CSP antibody served as a positive control for the sporozoite lysate. A polyclonal anti-PbGAPDH antibody (red arrow) identifies a sporozoite GAPDH band with identical mobility to the anti-P39 band. **(B)** Antibodies against each of the mimotope peptides (P39, P61, and P52) recognize bands with identical mobility as PbGAPDH (red arrow), but not the pET tag protein (blue arrow). Each panel shows two lanes, the left containing the pET tag protein alone and the right the tagged recombinant PbGAPDH protein. The antibodies used to probe the blots are indicated at the bottom of each panel. The anti-tag and anti-KLH antibodies served as a positive and negative controls, respectively. The anti-mouse GAPDH antibody was used as a positive control for identification of the GAPDH protein. The position of the recombinant protein and tag proteins is indicated by arrows to the right. All data are representative of two independent experiments. **(C)** Each anti-peptide antibody specifically recognizes its own peptide. The biotinylated peptide indicated at the top of each panel was bound to the wells of streptavidin-coated ELISA plates and tested for binding by the antibodies denoted at the bottom of each panel. No evidence of cross-reaction was detected. All data are representative of two independent assays.

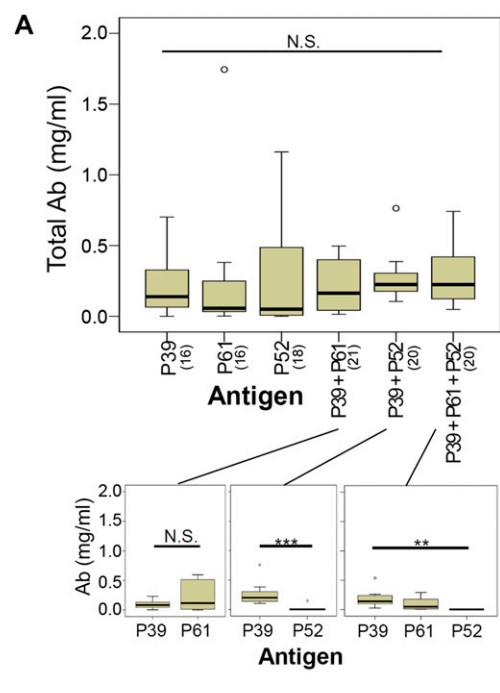

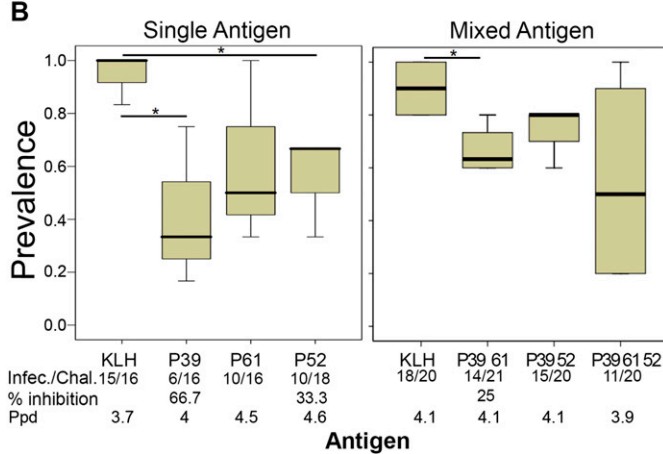

**Figure 2. Assessment of mimotope peptides as antigens for protection from *P berghei* infection.**
**(A)** Peptide immunogenicity. Mice were immunized either with a single antigen (50 µg of KLH-conjugated peptide), with a combination of two antigens (25 µg of each), or with a combination of three antigens (16.7 µg of each), as indicated. 1 wk after triple booster shots at 2-wk intervals, the amount of specific antibody produced by each mouse was determined with ELISA assays. Top panel: total antibody concentrations. The number of mice in each group is denoted in parenthesis. Bottom panels: specific antibody concentrations. All data are representative of two independent experiments. Error bars indicate SD. **(B)** Protection from infection by peptide immunization. Immunized mice were challenged by the bite of two infected *A stephensi* mosquitos. Prevalence of infection was determined with thin blood smears and Giemsa stain until 12 d post-infection. The number of infected mice/number of challenged mice and % inhibition are at the bottom of each panel. Data were pooled from three (single antigen) or four (mixed antigen) independent experiments. *P*-values (\*P < 0.05; \*\*P < 0.01; \*\*\*P < 0.001) were calculated using one-way ANOVA test (A) or Mann–Whitney *U* tests (B). Number of mice in each group in (A) is same as the number of mice in the corresponding group in (B). Ppd, average prepatent day.

by flow cytometry assays (Fig 3D). The histogram shows that the control, pET added, peak signal intensity is about 30 times stronger than that of the G6-treated specimen.

### Recombinant CD68 interacts with the PbGAPDH G6 fragment

Recombinant rat CD68 was expressed on the surface of human embryonic kidney 293T cells. Western blotting (Fig 4A), immuno-fluorescence (Fig 4B) and flow cytometry (Fig 4C) assays confirmed CD68 expression on the 293T cell surface. Either CD68-expressing 293T cells or untransfected control 293T cells were incubated with each sub-cloned PbGAPDH fragment. Fragment binding was determined by use of an anti-His tag antibody. Only the G6 fragment showed stronger binding affinity to CD68-expressing 293T cells than to the control cells (Fig 4D). ELISA assays were set up by attaching each recombinant PbGAPDH fragment to Ni-coated 96-well plates. Relative binding of recombinant CD68 from a lysate of CD68-expressing 293T cells (Figs 4A–C) to the fragments on the wells was determined with anti-CD68 antibody and secondary antibody. CD68 binding to the G6 fragment was significantly higher than to the pET control (Fig 4E). In addition, CD68-G6 fragment interaction was further confirmed with pull-down assays. Either pET, G3, or G6 fragment was incubated with CD68-expressing 293T cell lysate, and recombinant CD68 was pulled down with anti-CD68 antibody and protein-A agarose. The pulled down proteins were eluted and analyzed by using Western blotting (Fig 4F). As expected, pET did not bind to CD68 and the G6 fragment showed stronger binding to the recombinant CD68 than the G3 fragment. We conclude that G6 contains the major domain that mediates PGAPDH-CD68 interaction.

### Fine epitope mapping using a G6 peptide library identifies two epitopes recognized by the anti-P39 antibody

To identify the antigenic epitopes in PbGAPDH that are recognized by the anti-P39 antibody, we constructed an overlapping 15–amino acid peptide library that covers the G6 fragment (Fig 5A). Each biotinylated peptide was attached to a streptavidin-coated ELISA plate and tested for anti-P39 antibody binding (Fig 5B). Only peptides containing amino acids 1–20, and 41–60 were significantly recognized by the anti-P39 antibody. We mapped the two candidate epitope peptides, G6-1-20 (purple) and G6-41-60 (red), onto the homo tetrameric *P falciparum* GAPDH 3D structural model (Robien et al, 2006) using the Chimera software (Pettersen et al, 2004). As shown in Fig 5C, the two candidate epitope peptides closely contact each other on the surface of the molecule and are separated by the intervening peptide (orange) which is buried in the pocket (Video 1).

### Assessment of PbGAPDH epitope peptides as protective antigens for immunity against sporozoite liver infection

Mice were immunized with each of the two KLH-conjugated 20–amino acid–long epitope peptides, and antibodies were affinity purified using the biotinylated peptides attached to streptavidin-conjugated agarose beads. To test protective efficacy of anti-epitope peptide antibodies, 200 sporozoites mixed with 100 µg of each antibody were injected into each mouse. The same amount of the anti-KLH antibody and the anti-P39 antibody served as a negative and positive controls, respectively. As shown in Table 1, the two anti-epitope peptide antibodies showed similar protective efficacy as the anti-P39 antibody. Furthermore, we assessed the immunogenicity of the two peptides in mice. Immunofluorescence assays confirmed the

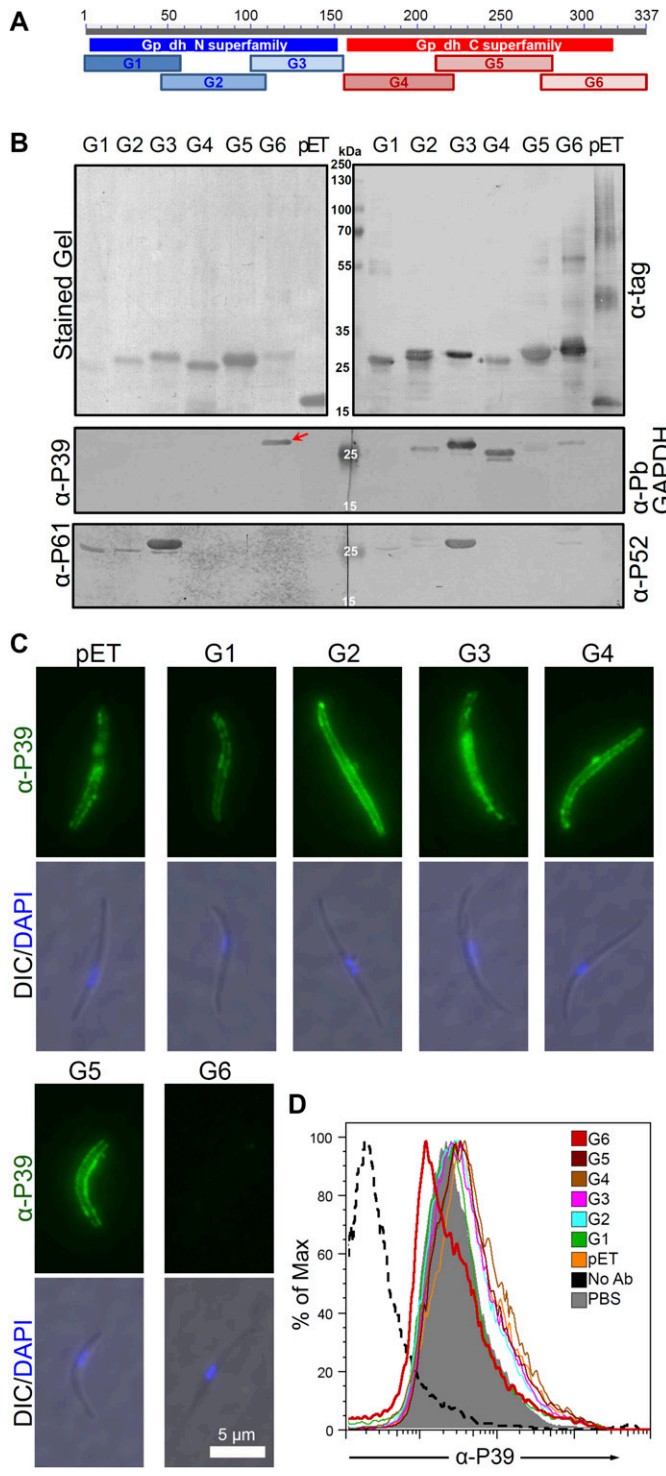

binding of the anti-epitope peptide antibodies to the sporozoite surface (Fig 6A). Western blot assays show that the antibodies specifically recognize PbGAPDH and do not cross-react with mammalian GAPDH (mGAPDH) (Fig 6B). Immunogenicity of each epitope peptide was determined by use of ELISA assays (Fig 6C) and the protective efficacy of each immunization was assessed by challenging immunized mice with the bite of two infected mosquitoes (Fig 6D). Immunization with the G6-1-20 and G6-41-60 peptide protected 30 and 50% of the mice from infection, respectively. Relatively milder protection by the G6-1-20 peptide immunization can be because of the weaker immunogenicity of the peptide (Fig 6C). In conclusion, data from Table 1 suggest that the efficacy of the antibodies against G6-1-20 and G6-41-60 are as protective as the anti-P39 antibody; however, our immunization regimens using KLH-conjugated linear peptide did not generate sufficient amount of antibody for effective targeting of sporozoite Kupffer cell traversal. Further optimization of the immunization regimen is required to achieve sterile immunity.

## Discussion

GAPDH plays a role in host cell invasion by several pathogenic microbes (Gil et al, 1999; Lama et al, 2009; Perez-Casal & Potter, 2016). In *Plasmodium*, surface-GAPDH has been reported in multiple stages, including in sporozoites (Lindner et al, 2013; Swearingen et al, 2016). The finding that *Plasmodium* surface-GAPDH is a sporozoite ligand for liver invasion (Cha et al, 2016) prompted us to explore the possibility that an antibody against PbGAPDH may interfere with infection of mammals by preventing Kupffer cell traversal and subsequent infection of the liver. Here, we show (i) that the anti-P39 antibody specifically recognizes the 65–amino acid C-terminal end of PbGAPDH (the G6 fragment) and (ii) that the G6 fragment can physically interact with the CD68 receptor. Mammalian cells also export GAPDH to their surface or secrete it into the extracellular milieu for iron homeostasis (Sheokand et al, 2014; Chauhan et al, 2015). Therefore, it is critical that a *Plasmodium* protective antibody does not cross-react with host GAPDH. When immunizing with the full-length PbGAPDH, the central region of the protein is immune-dominant, whereas antibodies against the major CD68 interaction domain are underrepresented. Thus, the most immunogenic epitopes may not be the most protective ones and identification of the relevant epitopes is critical for development of an effective vaccine. The anti-P39 antibody led to the identification of two peptides—G6-1-20 and G6-41-60—separated by a 20–amino acid intervening sequence. Mapping of these two anti-P39 antibody-binding epitopes on the *P falciparum* GAPDH 3D model place them in close proximity, making it likely that the P39 peptide mimics the junctional region of the two

**Figure 3. The anti-P39 antibody recognizes the C-terminal end of PbGAPDH.** **(A)** Diagrammatic representation of the recombinant PbGAPDH protein fragments covering the 337–amino acid full-length protein. **(B)** Purified recombinant PbGAPDH fragments described in (A) were separated by SDS–PAGE and transferred for Western blotting with anti-tag antibody as a loading control, with polyclonal antisera raised against the full length rPbGAPDH and with the anti-P39, anti-P61, and anti-P52 antibodies. The anti-P39 antibody specifically recognized the G6 fragment (red arrow). All data are representative of three independent experiments. **(C)** Only the G6 fragment competitively inhibits binding of an anti-P39 antibody to sporozoites. Each PbGAPDH fragment or the pET tag control

indicated at the top of each panel (200 µg/ml) was incubated overnight with *P berghei* sporozoites and 0.5% anti-P39 sera. Anti-P39 antibody binding to sporozoites was visualized with secondary antibody conjugated with Alexa-488. All data are representative of two independent immunofluorescence assays. **(D)** Experiment identical to that described in (C) but assayed by flow cytometry. The G6 fragment significantly inhibited anti-P39 antibody binding to the *P berghei* sporozoite surface.

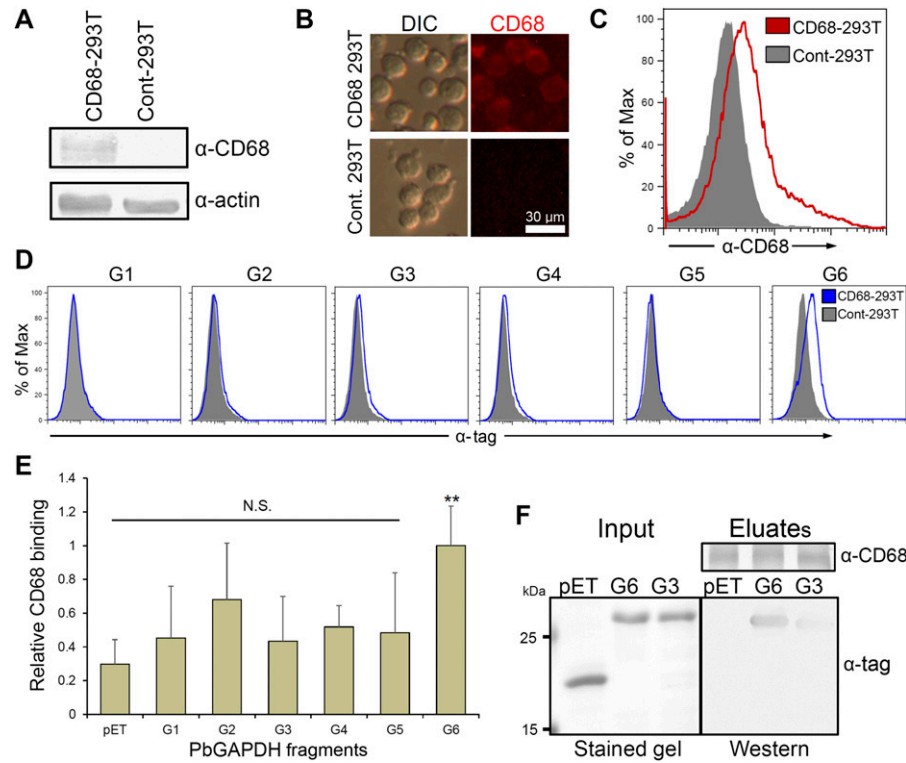

**Figure 4.  The PbGAPDH G6 fragment interacts with recombinant CD68.**
Either control human embryonic kidney cells (Cont-293T) or cells transfected to express rat CD68 protein on their surface (CD68-293T) were used. **(A)** Western blotting confirms CD68 expression by the recombinant, but not control cells. **(B)** Immunofluorescence assays with non-permeabilized cells show that the CD68 protein is on the surface of the recombinant cells. **(C)** Flow cytometry assays confirmed CD68 surface expression. (D) Each PbGAPDH sub-cloned fragment (c.f., Fig 2A) was incubated either with CD68-expressing 293T cells or untransfected control cells and analyzed by flow cytometry. PbGAPDH fragment binding was determined with an anti-His tag antibody. Only the G6 fragment detectably bound to the CD68 on the surface of recombinant cells. **(E)** Nickel-coated ELISA wells were incubated with individual PbGAPDH fragments (Fig 2A), and after triple washes, further incubated with a CD68-293T cell lysate. Recombinant CD68 binding to each sub-cloned PbGAPDH fragment was determined with anti-CD68 antibody. Binding of CD68 to the G6-coated well was significantly stronger than to the pET control. *P* value was calculated using one-way ANOVA test (**$P$ < 0.01). **(F)** Either the G6 fragment, the G3 fragment, or pET was incubated with a CD68-293T cell lysate, and the protein that bound to the CD68 was pulled down with an anti-CD68 antibody linked to protein-A agarose. Western blotting of the boiled-eluates determined that G6 bound more strongly to CD68 than G3. All data in this figure are representative of two or more independent experiments. Source data are available for this figure.

peptides. Similarly, a recent study identified two peptides separated by more than 20 amino acids as the epitope of a plasmodium falciparum CSP (PfCSP) C-terminus antibody and as is the case for PbGAPDH, the two PfCSP peptides are located next each other on the 3D conformational model (Scally et al, 2018). In this case, one PfCSP epitope is recognized by the heavy chain and the other is recognized by the light chain of a single antibody molecule. The two PbGAPDH epitope peptides were individually tested for binding to the recombinant CD68, however, no significant interaction was detected (data not shown). Possibly, an individual epitope peptide may not constitute a correct 3D structure for CD68 interaction. Our epitope mapping assays addressed the hypothesis that immunization with the cognate epitope (instead of an artificial peptide) may lead to more efficient infection interference. However, P39 has certain advantages as its amino acid sequence, DCAIVYAYDPCL, is totally unrelated with the host GAPDH protein and therefore has low possibility of eliciting auto-antibodies. Furthermore, it has similar protective properties as an antigen as the two PbGAPDH-derived peptides. Of note is that the P39 peptide was identified from a phage display screen of primary rat Kupffer cells and that the anti-P39 antibody inhibits rodent sporozoite–Kupffer cell interactions more efficiently (85%) than human sporozoite–macrophage interactions (56%) (see Fig 3B of Cha et al [2016]), suggesting that P39 may not optimally mimic the *P falciparum* epitopes. Thus, protection from infection by human *Plasmodium* parasites may require immunization with human parasite epitope antigens.

Our immunization regimen with P39 to target the CD68-dependent Kupffer cell traversal pathway confers 80% protection against challenge with single mosquito bites (Cha et al, 2016) and 67% protection against challenge with two mosquito bites (Fig 1D). Because

*Plasmodium* sporozoites have an alternate path to exit from the sinusoid by traversal of endothelial cells (Tavares et al, 2013), an antibody that inhibits the CD68-dependent sporozoite liver infection is not likely to generate complete sterile protection against sporozoite challenge. However, pGAPDH antigens are envisioned to be strong candidates to supplement the incomplete protection conferred by the most advanced malaria vaccine, RTS, S (Tran et al, 2015). Recently Luo et al (2017) reported that immunization of the recombinant CSP fused with macrophage inflammatory protein 3 *α* (MIP3*α*), immature dendritic cell–targeting chemokine, significantly improved antibody titration and duration of production. Fusion of pGAPDH epitope peptides with the MIP3*α*-CSP antigen will enable to test if pGAPDH-targeting antibody can supplement the moderate efficacy of RTS, S vaccine (Tran et al, 2015). The role of the anti–surface-GAPDH antibody in targeting other stages of *Plasmodium* parasite life cycle remains to be investigated.

# Materials and Methods

### Recombinant protein

Recombinant rat CD68, short-isoform, was generated as a membrane-bound protein in 293T cells (a human embryonic kidney cell line) as previously reported (Cha et al, 2015). Recombinant PbGAPDH fragment proteins were generated using pET32b expression system (Novagen). PbGAPDH fragments (fused to pET32b tag protein) or empty tag protein (thioredoxin-tag, N- and C-terminal His-tags, and S-tag) was purified with NiNTA agarose (QIAGEN).

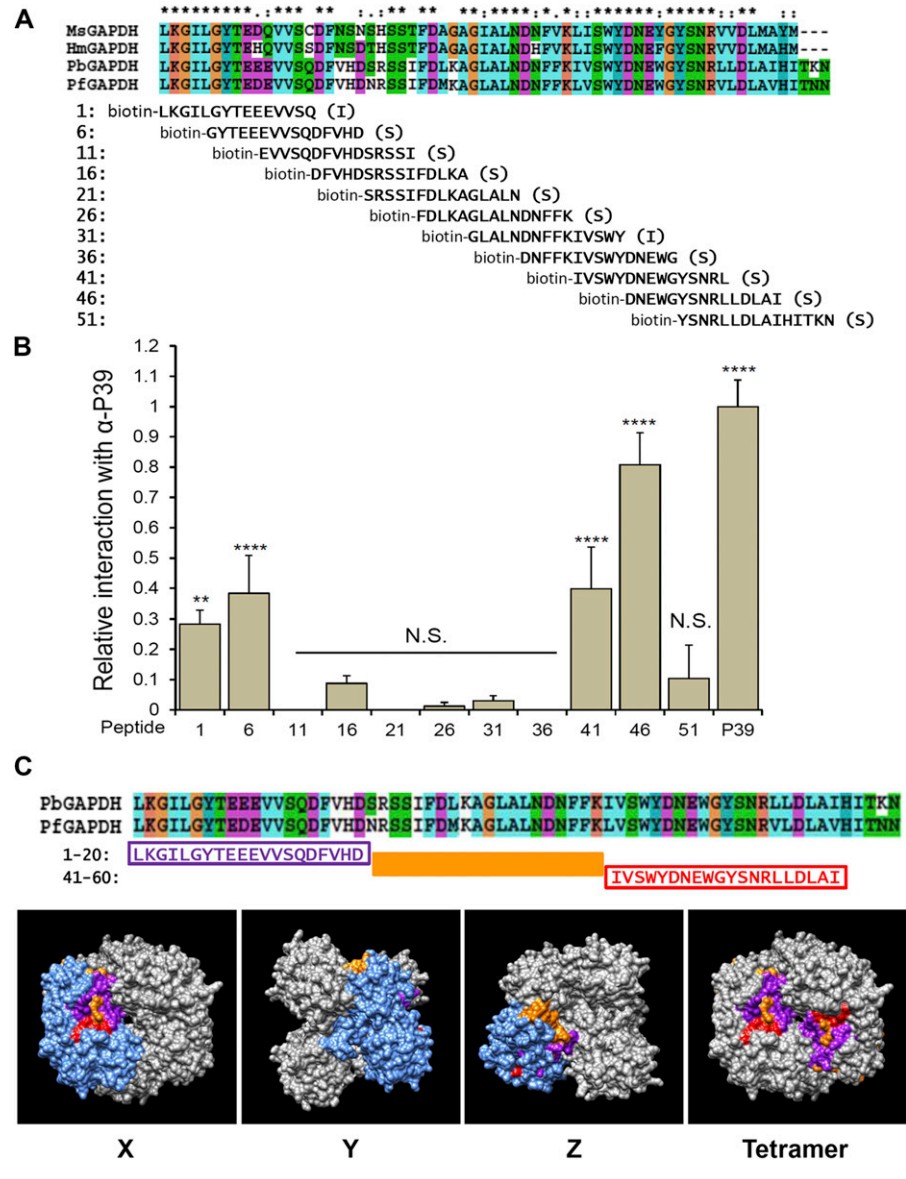

**Figure 5. Fine mapping assay with a G6 peptide library identifies two epitopes recognized by the anti-P39 antibody.**

**(A)** Eleven N-terminal biotinylated 15–amino acid–long peptides were synthesized to cover the 65 amino acids of the G6 fragment. The numbers to the left of each peptide denotes the position of the N-terminus within the G6 fragment. Predicted solubility is denoted in parenthesis as I, insoluble or S, soluble. Asterisks denote amino acids conserved among species, determined by multiple sequence alignment using Clustal X. **(B)** Each synthetic peptide was attached to a Ni-coated well of an ELISA plate and tested for anti-P39 antibody binding. Fragments containing amino acids 1–20 (fragments 1 and 6) and 41–60 (fragments 41 and 46) showed significant interaction with the anti-P39 antibody. $P$-values (**$P < 0.01$; ****$P < 0.0001$) were calculated using the one-way ANOVA test. Error bars indicate standard deviation. **(C)** Mapping of the two epitope peptides (1–20 and 41–60) on the 3D model of the homo-tetrameric PfGAPDH crystalized structure shows close proximity between the two epitope peptides. The upper panel shows alignment of PbGAPDH and *P falciparum* GAPDH (PfGAPDH) sequences. The G6-1-20 peptide is colored purple, the G6-41-60 peptide is colored red, and the intervening G6-21-40 peptide is colored orange. The lower panels show space-filling model with rotated views in the X, Y, and Z axes. Blue color represents the remaining amino acids in the monomeric PfGAPDH molecule. The two epitope peptides are attached to each other and exposed to the outside (X-axis), whereas the orange peptide is mostly buried inside the structure or exposed to the other side (Y- and Z-axes). Two sets of epitope peptides form a pocket-like structure in the tetrameric view; the other two sets are on the other side of the tetramer, not visible in this view.

## Immunization and challenge assays

For immunization, 21–25-g female Swiss Webster mice were injected with 50 μg per mouse of each KLH conjugated peptide or with KLH alone to serve as a control. For the multiple peptide immunization regimens, we used a mixture of the two peptides (25 μg each) or three peptides (16.7 μg each) per mouse. Priming with complete Freund's adjuvant was followed by triple boosts with incomplete Freund's adjuvant every other week. The immunization and challenge assay illustrated in Fig 6D was repeated with another adjuvant, AddaVax (InvivoGen), a squalene-based oil-in-water nanoemulsion with a formulation similar to MF59 that has been licensed in Europe for adjuvanted flu vaccines (O'Hagan, 2007). AddaVax has been reported to produce similar immunity as Freund's adjuvant (Hawksworth, 2017) and our result confirmed this (data not shown). Mouse antibody was titrated 1 wk after the final boost using

biotinylated peptide attached to streptavidin-coated ELISA plates (Thermo Fisher Scientific). Purified antibody was used for the standard curve assay in each ELISA assay. 10 d after the last boost, immunized mice were challenged with *P berghei* sporozoites with biting by two infected *A stephensi* mosquitoes. For mouse challenge assays, highly infected female mosquitoes were selected by examination of salivary gland fluorescence, 18 d after mosquito feeding on mice infected with tdTomato-expressing parasites (Graewe et al, 2009). Mouse parasite infection was determined by thin blood smears followed by Giemsa staining from days 4 to 12 post-challenge.

## Antibodies

Polyclonal anti-PbGAPDH antisera were generated by immunization of mice, as described in the previous section, with 50 μg per mouse

**Table 1. Anti-PbGAPDH epitope antibodies inhibit sporozoite liver invasion with similar efficacy as anti-P39 antibody.**

| Ab | α-KLH | α-P39 | α-G6-1-20 | α-G6-41-60 |
|---|---|---|---|---|
| Exp 1 | 5/5 | 0/5 | 0/5 | 1/5 |
| Exp 2 | 4/5 | 2/5 | 0/5 | 0/5 |
| Total | 9/10 | 2/10 | 0/10 | 1/10 |
| % inhibition | — | 77.8 | 100 | 88.9 |

A total of 200 *P berghei* salivary gland sporozoites together with 100 μg of each antibody were intravenously injected into each mouse. Prevalence of infection was determined with thin blood smears and Giemsa stain up to day 12 post-injection. % inhibition calculated relative to the α-KLH control.

of full-length recombinant PbGAPDH protein with C-terminal E-tag and with no N-terminal tags after enterokinase treatment, as described previously (Cha et al, 2016). Therefore, antisera from

immunization with recombinant PbGAPDH with C-terminal E-tag did not cross-react with the empty tag protein, pET, which contains thioredoxin-tag, His-tag, and S-tag (Fig 2B). All anti-synthetic peptide antibodies were purified with the corresponding biotinylated antigen peptides bound to streptavidin agarose beads (Millipore). Purified antibodies were stored in PBS and the concentration was determined with Western blotting assays with reference of a commercially available antibody.

## PbGAPDH-CD68 interaction measurements with ELISA and pull-down assays

Recombinant PbGAPDH fragment or pET32b tag protein was attached to a Ni-coated ELISA plate (Thermo Fisher Scientific). After washing and blocking with 1% BSA in PBS, the coated wells were further incubated with lysate of 293T7 cells which were

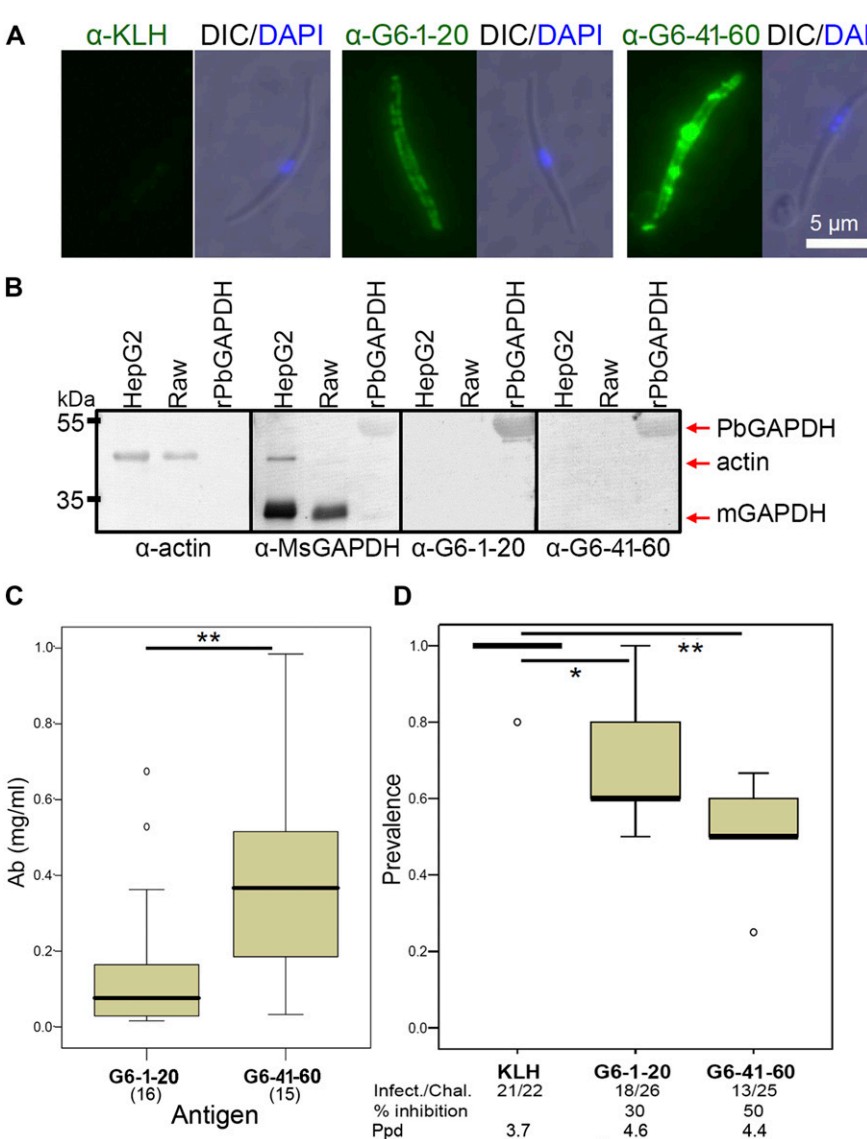

**Figure 6. Antibodies against PbGAPDH epitope peptides bind to sporozoite surface-GAPDH and inhibit sporozoite liver infection.**
Mice were immunized with the KLH-conjugated epitope peptides G6-1-20, G6-41-60, or with KLH only as a negative control (c.f., Fig 5). **(A)** Immunofluorescence assays verified that the anti-PbGAPDH epitope peptide antibodies bind to the surface of non-permeabilized *P berghei* sporozoites. **(B)** Anti-PbGAPDH epitope antibodies specifically recognize parasite GAPDH, not mGAPDH. Commercial anti-mouse GAPDH antibody (α-MsGAPDH) served as a positive control to stain mGAPDH from a HepG2 (a human hepatoma cell line) lysate, from a Raw cell (a mouse macrophage-like cell line) lysate, and recombinant PbGAPDH protein. Anti-actin antibody was used as a loading control. Red arrows indicate the band of *P berghei* GAPDH, actin, and mouse GAPDH, respectively. **(C)** Comparison of the immunogenicity of two PbGAPDH epitope peptides, G6-1-20 and G6-41-60. Immunization with G6-41-60 produced more antibody than G6-1-20 immunization. Data pooled from three independent experiments. Number of assayed mice denoted in parenthesis at the bottom of the panel. **(D)** Immunization with epitope peptides protects mice from infection. Immunized mice were challenged by the bites of two infected *A stephensi* mosquitos. Prevalence of mouse infection was determined by thin-blood smears and Giemsa stain until day 12 post-infection. Infect./Chal.: number infected over number of challenged mice. Ppd: average prepatent days. Data pooled from three independent experiments with Freund's adjuvant and two independent experiments with AddaVax adjuvant which showed similar results as Freund's adjuvant. *P*-values (\**P* < 0.05; \*\**P* < 0.01) were calculated using the one-way ANOVA test (C) or Mann–Whitney *U* tests (D). Error bars in (C) indicate SD.

transfected to express rat CD68. An anti-S tag antibody determined the amount of PbGAPDH fragments or pET32b tag protein attached to the wells. The amount of CD68 binding to the attached protein was determined with an anti-rat CD68 antibody (AbD Serotec) and normalized to the amount of attached protein. For the pull-down assays, recombinant rat CD68 was bound to protein-A beads (Invitrogen) using anti-rat CD68 antibody (AbD Serotec). After incubation with recombinant PbGAPDH fragments or pET32b tag protein, the CD68-Protein A beads were boiled in Laemmli buffer to elute bound proteins. Western blotting assays with an anti-thioredoxin tag antibody visualized the recombinant PbGAPDH G3 and G6 fragments and pET32b tag proteins.

### Fine epitope mapping assay with ELISA and a peptide library

A 15–amino acid N-terminal biotinylated synthetic peptide library was generated (Peptide 2.0). The biotinylated peptide was attached to a streptavidin-coated 96-well plate and binding of anti-P39 antibodies were visualized with secondary antibody. Among peptide groups showing no significant binding to the anti-P39 antibody, peptide-51 showed the highest reactivity with anti-P39 antibody and served as a control for calculation of $P$-values with peptide 1, 6, 41, and 46.

### Immunofluorescence and flow cytometry assays

For binding competition assays of anti-P39 antibody with PbGAPDH fragments, *P berghei* sporozoites were harvested and fixed in PBS with 4% paraformaldehyde for 30 min. Fixed sporozoites were incubated overnight in PBS with 200 $\mu$g/ml of each recombinant PbGAPDH fragment and 0.5% anti-P39 sera. Binding of anti-P39 antibody was visualized with Alexa Fluor 488 (green)-conjugated secondary antibody (Invitrogen). Flow cytometry data were acquired with a FACSCalibur apparatus (BD Biosciences) and analyzed using the FlowJo v8.7 software (Tree Star, Inc.). For immunofluorescence assays, nuclei were stained with DAPI (blue).

### Peptide mapping on the homo tetrameric PfGAPDH 3D model using Chimera software

Chimera software was obtained from http://www.rbvi.ucsf.edu/chimera and analysis was followed by user's guide (Pettersen et al, 2004). PDB file of the PfGAPDH crystal structure was obtained from 2B4R.

## Statistics for data analysis

ELISA assays were analyzed using the one-way ANOVA test. Nonparametric comparisons between groups were performed using the Mann–Whitney $U$ test.

## Supplementary Information

## Acknowledgements

This work was supported by grant AI123613 from the National Institute of Allergy and Infectious Diseases. We thank the Johns Hopkins Malaria Research Institute mosquito and parasite core facilities for help with mosquito rearing and with *P falciparum* gametocyte culture. Support from the Johns Hopkins Malaria Research Institute and the Bloomberg Philanthropies is gratefully acknowledged. Supply of human blood was supported by the National Institutes of Health grant RR00052. Molecular graphics and analyses were performed with the UCSF Chimera package. Chimera is developed by the Resource for Biocomputing, Visualization, and Informatics at the University of California, San Francisco (supported by NIGMS P41-GM103311).

### Author Contributions

SJ Cha: conceptualization, data curation, formal analysis, funding acquisition, validation, investigation, methodology, and writing—original draft, review, and editing.
KJ MacLean: data curation, software, and formal analysis.
M Jacobs-Lorena: conceptualization, supervision, funding acquisition, investigation, project administration, and writing—original draft, review, and editing.

### Conflict of Interest Statement

The authors declare that they have no conflict of interest.

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
