## [Reviewer comments · Life Science Alliance]

Life Science Alliance

IDENTIFICATION OF PLASMODIUM GAPDH EPITOPES FOR GENERATING ANTIBODIES THAT INHIBIT MALARIA INFECTION

Sung-Jae Cha, Kyle MacLean, and Marcelo Jacobs-Lorena

DOI: 10.26508/lsa.201800111

Corresponding author(s): Marcelo Jacobs-Lorena, Johns Hopkins Bloomberg School of Public Health

Review Timeline:

Submission Date:	2018-06-19
Editorial Decision:	2018-07-20
Revision Received:	2018-08-15
Editorial Decision:	2018-08-31
Revision Received:	2018-09-05
Accepted:	2018-09-07

Scientific Editor: Dr. Andrea Leibfried

Transaction Report:

July 20, 2018

Re: Life Science Alliance manuscript #LSA-2018-00111-T

Dr. Marcelo Jacobs-Lorena
Johns Hopkins Bloomberg School of Public Health
Molecular Microbiology & Immunology
615 N. Wolfe Street
615 N. Wolfe St., E5132
Baltimore, MD 21205

Dear Dr. Jacobs-Lorena,

Thank you for submitting your manuscript entitled "IDENTIFICATION OF PLASMODIUM GAPDH VACCINE ANTIGENS THAT TARGET SPOROZOITE LIVER INVASION" to Life Science Alliance. The manuscript was assessed by expert reviewers, whose comments are appended to this letter.

As you will see, the reviewers appreciate your work. They raise a few concerns, and their revision requests either relate to missing controls or aim at strengthening your main conclusions. We would thus be happy to publish a revised version of your work in Life Science Alliance, addressing the concerns of the three reviewers.

-- High-resolution figure, supplementary figure and video files uploaded as individual files: See our detailed guidelines for preparing your production-ready images, <http://life-science-alliance.org/authorguide>

B. MANUSCRIPT ORGANIZATION AND FORMATTING:

Full guidelines are available on our Instructions for Authors page, <http://life-science-alliance.org/authorguide>

Thank you for this interesting contribution to Life Science Alliance. We are looking forward to receiving your revised manuscript.

Sincerely,

Andrea Leibfried, PhD
Executive Editor
Life Science Alliance
Meyerohofstr. 1
69117 Heidelberg, Germany
t +49 6221 8891 502
e a.leibfried@life-science-alliance.org
www.life-science-alliance.org

Reviewer #1 (Comments to the Authors (Required)):

The authors previously identified CD68 as a Kupffer cell receptor involved in sporozoite invasion (J Exp Med. 2015 Aug 24;212(9):1391-403) and reported that parasite GAPDH is a parasite ligand binding to CD68 and that antibodies against a mimotope peptide (P39) block sporozoite invasion, making it a prehepatic malaria vaccine candidate (J Exp Med. 2016 Sep 19;213(10):2099-112).

Here, Cha and colleagues report a follow-up study, where they determined that the C-terminus of Plasmodium GAPDH is responsible for the interaction with CD68, and that two peptides within the GAPDH C-terminus act as protective antigens and are candidates for a pre-erythrocytic malaria vaccine. The paper is well written and the conclusions generally supported by the data. However, there are a number of points that should be addressed, as detailed below, to strengthen the manuscript.

- 1) In this and previous studies, the authors used a non-stringent challenge approach. They used Swiss Webster mice and low numbers of parasites (2 mosquito bites or less than 1000 sporozoites). This is not the most relevant system, as reflected by the fact that some of the control mice do not develop parasitemia after challenge (Fig1D and Table 1). They should perform experiments in C57BL/6 mice, which are highly susceptible to *P. berghei* sporozoites and provide a more relevant system to assess vaccine efficacy.
- 2) In Table 1, the authors show a very good protective efficacy of purified antibodies (obtained after peptide immunization), which contrasts with the low protective efficacy in peptide-immunized mice (Fig 5D). How do the authors explain this discrepancy? What is the amount of anti-peptide antibodies achieved after peptide immunization in this model?
- 3) The title should be changed. First, vaccine antigens do not target sporozoite invasion, vaccine candidates may do so. Second, the title should reflect more accurately the content of the study (i.e. the identification of candidate peptides in the C-terminus of GAPDH) and the novelty as compared with the two previous publications of the same group.
- 4) In several figures (1A, 2B), the blots are cut just above the band of interest (55kDa). It would be more appropriate to display a larger part of the blots.
- 5) The authors conclude from figure 1B that "Each peptide appears to mimic a different PbGAPDH epitope". Actually, fig 1B only shows that antibodies against each peptide only recognize this peptide and not the others. It would be interesting to test the anti-P61 and anti-P52 antibodies against GAPDH fragments, as shown for anti-P39 in Fig 2B. This would delimit more precisely the specificity of the different anti-mimotopes.
- 6) Fig2C shows IFA signal in FACS (albeit reduced) but not microscopy. How do the authors explain this discrepancy?
- 7) Fig3F shows that G3 binds to CD68, which is not consistent with the authors conclusion regarding the binding region of GAPDH. This needs to be clarified.
- 8) At the end of the results section, the authors indicate that "Antibodies to G6-1-20 and G6-41-60 are highly protective although our immunization regimens did not generate sufficient amount of antibody for effective targeting of sporozoite Kupffer cell traversal. What do the authors mean?
- 9) In the discussion, a sentence should be completed ("Thus, protection from infection by human Plasmodium parasites may require immunization with...")

Reviewer #2 (Comments to the Authors (Required)):

The paper reports on the use of two antibodies raised peptides of Plasmodium GAPDH as inhibitors of malarial parasite entry into the liver.

The message of the paper would improve if the authors would test whether/how their antibodies could synergistically act with anti-CSP antibodies over a range of concentrations that include a reasonable range for human use (currently the 100 ug used to inject mice (3 ml of blood) correspond to the injection of 200 mg of antibody into a human - 5000 ml of blood) by i.v. injection followed by mosquito bite and i.v. injection of sporozoites. Injection at the same time seems a suboptimal assay but antibodies should be allowed to circulate for some time to mimic possible use as a vaccine.

Minor issues:

Avoid using 'vaccine' in title

Intro: Tavares et al showed that half the sporozoites do not pass through Kupffer cells. This has to be clearly introduced and the rationale has to be re-balanced accordingly. Just mentioning this paper almost in passing in the conclusion is borderline dishonest. Same line of thought: the idea of complementing the existing non-perfect RTS/S vaccine should be put into introduction.

The languages and avoid statements like 'elicits strong protective immunity', when it is at best 'partial immunity'

Please don't report data too precisely, inhibition of e.g. 84,7 vs 56,3 % should be 85 vs 56%.

Reviewer #3 (Comments to the Authors (Required)):

The manuscript entitled « Identification of Plasmodium GAPDH vaccine antigens that target sporozoite liver invasion » by Cha and collaborators documents the identification of two 20-mer amino acid sequences from PbGAPDH that when conjugated to KLH and used with CFA/IFA to immunize Swiss mice (4 immunizing doses), passively and/or actively protected mice against sporozoite infection. In addition they show the binding of region G6 of GAPDH to CD68, but not of the two protective fragments G6-1-20 and G6-41-60 (data not shown).

I have a few comments about interpretation of results and experimental data.

1- The title is misleading since no experiment showing inhibition of liver invasion (crossing of the sinusoidal barrier) is presented. In this manuscript, all readouts are related to blood infection (prevalence). In the previous manuscripts, the authors used "entry?" in macrophages as a proxy of cellular transmigration. Anti-P39, P61 and P52 antibodies inhibit "entry" but increase attachment to macrophage (Fig. 2B, Cha et al, JEM 2016).

On the other hand, P39, P61 and P52, inhibited both entry and attachment (Fig. 2AB, Cha et al, JEM 2015), indicating that the mechanism of protection could be other than liver invasion, eg, increased phagocytosis via Fc receptor.

Moreover, what the authors call "entry" in their assay is a measure of intracellular sporozoites, i.e.,

the total of sporozoites that invade macrophages plus those that were phagocytosed. However, since macrophages can kill intracellular sporozoites the readout at 2h reflects intracellular survivors. Importantly, this is a non-physiological assay since at 2h no substantial numbers of sporozoites are found inside/ traversing Kupffer cells in vivo. Please perform an experiment showing that immunization using GAPDH inhibits sporozoite liver invasion (translocation to the hepatic parenchyma) or change the title.

2- The introduction is categorical, however the cited references do not sustain the authors' statements. For example, the paper of Cerami et al, 1994 shows that CSP binds in vivo to the microvilli of hepatocytes (Please see also Cerami et al, Cell 1992) and not to stellate cells.

3- The next sentence, "Sporozoites leave the circulation mainly by traversing Kupffer cells..." is referenced by 6 manuscripts. Out of this 6, only the paper of Frevert et al, Plos Biol 2005 quantified the dynamic process of sinusoidal extravasation, but without the spatial resolution or appropriate markers to determine whether sporozoites were traversing Kupffer cells (KC) during this translocation. More recently, Tavares et al, JEM 2013 using a method to detect KC traversal by dynamic imaging showed that at least around 25% of estimated extravasation events are related to direct traversal of KC. This paper is much more relevant than the last cited manuscript (Sturm et al, Science 2006), which refers to the intravasation of merozoites, not to the extravasation of sporozoites. Please correct these two sentences (stellate cells and KC traversal).

4- I'm surprised by the fact that the authors are using CFA as adjuvant (Grumpstrup-Scott J, Greenhouse DD. 1988. NIH intramural recommendation for the research use of complete Freund's adjuvant. ILAR News 20:9). Was this adjuvant chosen because the fragments of PbGAPDH present low immunogenicity? The protocol of immunization/challenge should be repeated with an adjuvant that could be potentially used in humans (timeframe: ~1.5 mo)

5- Control using sporozoites in the WB of figures 1A (α-P39) and 5B (α-fragments of GAPDH) are missing. This is important because antibodies against P39 recognize another band (70-100 kD) in sporozoites (Fig. 1A; Cha et al, JEM 2016))

In addition, the loop sequence between the two cysteines of P39 matches almost perfectly a fragment of PBANKA_1317200 and only partially the region of peptide 41 (GAPDH, Fig. 4A)

```
> PBANKA_1317200.1-p1 | transcript=PBANKA_1317200.1 | gene=PBANKA_1317200 |  
organism=Plasmodium_berghei_ANKA | gene_product=NAD(P) transhydrogenase, putative |  
transcript_product=NAD(P) transhydrogenase, putative | location=PbANKA_13_v3:722581-  
726493(-) | protein_length=1201 | sequence_SO=chromosome | SO=protein_coding |  
is_pseudo=false Length=1201 Score = 20.8 bits (42), Expect = 6.0, Method: Composition-based  
stats. Identities = 7/8 (88%), Positives = 8/8 (100%), Gaps = 0/8 (0%)
```

```
Query 1 CAMYAYDPC 8  
+IVYAYDP  
Sbjct 863 SIVYAYDP 870
```

P39 1 -----CAIYAYDPC----- 10

||::||:

PbGAPDH 301 KAGLALNDNFFKIVSWYDNEWGYSNRLLDLAIHITKN 337

6- It would be important also to perform the experiments of figure 2 using antibodies against P61 and P52. Albeit less protective than P39, they could target the N-terminal domain of GAPDH, which is much less conserved than the C-terminal domain when comparing the *P.falciparum* and human sequences, or *P.berghei* and rodent sequences. Timeframe (2 weeks).

7- Pre-patent period should be included in the figures 1D, 5C and table I to assess the level of control infection.

RESPONSE TO THE REVIEWER'S COMMENTS

We thank the reviewers for their time reading our manuscript and for their insightful comments. Our answers here and the changes in the main text, are in blue font.

Reviewer #1 (Comments to the Authors (Required)):

The authors previously identified CD68 as a Kupffer cell receptor involved in sporozoite invasion (J Exp Med. 2015 Aug 24;212(9):1391-403) and reported that parasite GAPDH is a parasite ligand binding to CD68 and that antibodies against a mimotope peptide (P39) block sporozoite invasion, making it a prehepatic malaria vaccine candidate (J Exp Med. 2016 Sep 19;213(10):2099-112).

Here, Cha and colleagues report a follow-up study, where they determined that the C-terminus of Plasmodium GAPDH is responsible for the interaction with CD68, and that two peptides within the GAPDH C-terminus act as protective antigens and are candidates for a pre-erythrocytic malaria vaccine. The paper is well written and the conclusions generally supported by the data. However, there are a number of points that should be addressed, as detailed below, to strengthen the manuscript.

1) In this and previous studies, the authors used a non-stringent challenge approach. They used Swiss Webster mice and low numbers of parasites (2 mosquito bites or less than 1000 sporozoites). This is not the most relevant system, as reflected by the fact that some of the control mice do not develop parasitemia after challenge (Fig1D and Table 1). They should perform experiments in C57BL/6 mice, which are highly susceptible to *P. berghei* sporozoites and provide a more relevant system to assess vaccine efficacy.

We think that using Swiss Webster, an outbred strain of diverse genetic background is preferable to use inbred C57BL/6 mouse, as this approaches more closely the situation in the field where humans are genetically highly heterogeneous. Furthermore, challenging the mice with two mosquito bites also closely approaches the situation in the field, where *infected* mosquitoes are rare even in high-transmission areas and humans are not usually bitten by more than one infected mosquito bite. If we were to over-challenge the mice with an artificially large number of sporozoites, the assay would lose sensitivity as even a decrease in the success of sporozoite infection of the liver would not be reflected in a decrease of mice being infected (prevalence). Finally, the aim of the study is identification of the protective epitopes on the PbGAPDH and not assessment of vaccine efficacy.

2) In Table 1, the authors show a very good protective efficacy of purified antibodies (obtained after peptide immunization), which contrasts with the low protective efficacy in peptide-immunized mice (Fig 5D). How do the authors explain this discrepancy? What is the amount of anti-peptide antibodies achieved after peptide immunization in this model?

Table 1 show that the antibodies are highly protective while in mice immunization experiments the protection was lower. We attribute this difference to the less-than-ideal immunogenicity of the KLH-conjugated peptides. This interpretation is mentioned at the end of the Results section, when we state that “*data from Table 1 suggest that the efficacy of the antibodies against G6-1-20 and G6-41-60 are as protective as the anti-P39 antibody, however our immunization regimens using KLH-conjugated linear peptide did not generate sufficient amount of antibody for effective targeting of sporozoite Kupffer cell traversal. Further optimization of the immunization regimen is required to achieve sterile immunity*”. A possible approach to improve immunogenicity is suggested at the end of the Discussion, where we suggest fusing the peptides with the macrophage inflammatory protein 3 alpha (MIP3 α).

3) The title should be changed. First, vaccine antigens do not target sporozoite invasion, vaccine candidates may do so. Second, the title should reflect more accurately the content of the study (i.e. the identification of candidate peptides in the C-terminus of GAPDH) and the novelty as compared with the two previous publications of the same group.

Thank you for this suggestion. The title was changed.

4) In several figures (1A, 2B), the blots are cut just above the band of interest (55kDa). It would be more appropriate to display a larger part of the blots.

Fig. 1A & Fig 2B show purified recombinant proteins. As shown in the new Figures, stained gels after purification clarify that cropped images are representative of the claimed results.

5) The authors conclude from figure 1B that "Each peptide appears to mimic a different PbGAPDH epitope". Actually, fig 1B only shows that antibodies against each peptide only recognize this peptide and not the others. It would be interesting to test the anti-P61 and anti-P52 antibodies against GAPDH fragments, as shown for anti-P39 in Fig 2B. This would delimit more precisely the specificity of the different anti-mimotopes.

This is a good suggestion. The new Fig. 3B show the results of such experiment.

6) Fig2C shows IFA signal in FACS (albeit reduced) but not microscopy. How do the authors explain this discrepancy?

We note that in the histogram of the new Fig.3D the fluorescence intensity on the x-axis is displayed on a log scale. In this figure, the signal intensity after addition of the control pET peptide is about 30-fold stronger than the signal after addition of the G6 peptide. The breadth of signal detection of the IFA technique is much narrower than that of flow cytometry.

7) Fig3F shows that G3 binds to CD68, which is not consistent with the authors conclusion

regarding the binding region of GAPDH. This needs to be clarified.

The G3 signal in the original Fig. 3F (new Fig. 4F) is much weaker than that of G6. Additional assays (new Fig. 3B) showed P61 and P52 mimic epitopes on the G3 fragment, which also implies G3 has CD68 binding domain with weaker affinity than G6. These conclusions are corroborated by data in new Figs. 4D and 4E. Of note, anti-P39 antibody only binds to G6.

8) At the end of the results section, the authors indicate that "Antibodies to G6-1-20 and G6-41-60 are highly protective although our immunization regimens did not generate sufficient amount of antibody for effective targeting of sporozoite Kupffer cell traversal. What do the authors mean?"

What we meant to convey is that the data of Table 1 show that the antibodies against epitope peptides inhibit sporozoite liver invasion efficiently when present in sufficient amounts (100 µg of affinity-purified antibody was used), however immunization of the KLH conjugated peptide did not produce enough antibody to saturate the sporozoite ligand. We made changes in the text that hopefully clarify this issue. See also our comments provided in paragraph 2).

9) In the discussion, a sentence should be completed ("Thus, protection from infection by human Plasmodium parasites may require immunization with...")

This was done.

Reviewer #2 (Comments to the Authors (Required)):

The paper reports on the use of two antibodies raised peptides of Plasmodium GAPDH as inhibitors of malarial parasite entry into the liver.

The message of the paper would improve if the authors would test whether/how their antibodies could synergistically act with anti-CSP antibodies over a range of concentrations that include a reasonable range for human use (currently the 100 µg used to inject mice (3 ml of blood) correspond to the injection of 200 mg of antibody into a human - 5000 ml of blood) by i.v. injection followed by mosquito bite and i.v. injection of sporozoites. Injection at the same time seems a suboptimal assay but antibodies should be allowed to circulate for some time to mimic possible use as a vaccine.

Thank you for this suggestion. Our manuscript reports on the initial aim of characterizing PbGAPDH antigens. We are presently pursuing co-immunization with PbGAPDH and CSP antigens for possible synergistic protective efficacy. We comment on this topic at the end of the Discussion of the revised manuscript.

Injection antibody and sporozoite at the same time was due to the limited amount of the affinity-purified antibody. The point here is to show the ability of the antibody to inhibit

infection, if it is present in sufficient concentration. Please also see comments in our response to Reviewer 1, paragraph 2).

Minor issues:

Avoid using 'vaccine' in title

This is a good suggestion. The title was changed accordingly.

Intro: Tavares et al showed that half the sporozoites do not pass through Kupffer cells. This has to be clearly introduced and the rationale has to be re-balanced accordingly. Just mentioning this paper almost in passing in the conclusion is borderline dishonest. Same line of thought: the idea of complementing the existing non-perfect RTS/S vaccine should be put into introduction.

We did. Tavares et al. is now cited in the introduction.

The languages and avoid statements like 'elicits strong protective immunity', when it is at best 'partial immunity'. Please don't report data too precisely, inhibition of e.g. 84,7 vs 56,3 % should be 85 vs 56%.

Thank you for this suggestion. This was changed.

Reviewer #3 (Comments to the Authors (Required)):

The manuscript entitled « Identification of Plasmodium GAPDH vaccine antigens that target sporozoite liver invasion » by Cha and collaborators documents the identification of two 20-mer amino acid sequences from PbGAPDH that when conjugated to KLH and used with CFA/IFA to immunize Swiss mice (4 immunizing doses), passively and/or actively protected mice against sporozoite infection. In addition, they show the binding of region G6 of GAPDH to CD68, but not of the two protective fragments G6-1-20 and G6-41-60 (data not shown).

I have a few comments about interpretation of results and experimental data.

1- The title is misleading since no experiment showing inhibition of liver invasion (crossing of the sinusoidal barrier) is presented. In this manuscript, all readouts are related to blood infection (prevalence). In the previous manuscripts, the authors used "entry?" in macrophages as a proxy of cellular transmigration. Anti-P39, P61 and P52 antibodies inhibit "entry" but increase attachment to macrophage (Fig. 2B, Cha et al, JEM 2016). On the other hand, P39, P61 and P52, inhibited both entry and attachment (Fig. 2AB, Cha et al , JEM 2015),

indicating that the mechanism of protection could be other than liver invasion, eg, increased phagocytosis via Fc receptor. Moreover, what the authors call "entry" in their assay is a measure of intracellular sporozoites, i.e., the total of sporozoites that invade macrophages plus those that were phagocytosed. However, since macrophages can kill intracellular sporozoites the readout at 2h reflects intracellular survivors. Importantly, this is a non-physiological assay since at 2h no substantial numbers of sporozoites are found inside/traversing Kupffer cells in vivo. Please perform an experiment showing that immunization using GAPDH inhibits sporozoite liver invasion (translocation to the hepatic parenchyma) or change the title.

Mimotope peptides bind to the Kupffer cell CD68, and the anti-P39 antibody binds to the sporozoite surface GAPDH. Incubation of the mimotope peptide at 0.4 mg/ml concentration may activate CD68 scavenging machinery for peptide uptake, which may competitively inhibit sporozoite attachment and entry into Kupffer cells. On the other hand, antibody binding to sporozoite surface does not inhibit CD68 machinery, however antibody binding to the sporozoite surface GAPDH epitope may inhibit interaction with CD68 by directing to the Fc receptor, which may lead sporozoite not to initiate Kupffer cell entry. Fig. 2B of Cha et al., 2016 shows that antibody binding to sporozoite GAPDH did not activate phagocytosis via Fc receptor. If it did, sporozoite attachment in the anti-mimotope antibody-treated groups should have decreased as compared to the control antibody-treated group (Fig. 2B, right panel). In any case, any decrease of Kupffer cell traversal will result in decrease of liver invasion. The main aim of this work was to identify protective GAPDH epitopes. Neither the title nor the experiments in this manuscript address the detailed mechanism of decreased infection.

2- The introduction is categorical, however the cited references do not sustain the authors' statements. For example, the paper of Cerami et al, 1994 shows that CSP binds in vivo to the microvilli of hepatocytes (Please see also Cerami et al, Cell 1992) and not to stellate cells.

Thank you for pointing this out. This was changed.

3- The next sentence, "Sporozoites leave the circulation mainly by traversing Kupffer cells..." is referenced by 6 manuscripts. Out of this 6, only the paper of Frevert et al, Plos Biol 2005 quantified the dynamic process of sinusoidal extravasation, but without the spatial resolution or appropriate markers to determine whether sporozoites were traversing Kupffer cells (KC) during this translocation. More recently, Tavares et al, JEM 2013 using a method to detect KC traversal by dynamic imaging showed that at least around 25% of estimated extravasation events are related to direct traversal of KC. This paper is much more relevant than the last cited manuscript (Sturm et al, Science 2006), which refers to the intravasation of merozoites, not to the extravasation of sporozoites. Please correct these two sentences (stellate cells and KC traversal).

We specified references in more detail in the text. Of note, Frevert et al. (2005) and Tavares et al. (2013) used transgenic mouse strains that express GFP in the endothelial cell lining. Tavares et al. used F4/80 antibody to detect Kupffer cell using intravital confocal microscopy. However not all Kupffer cells are detected by the F4/80 antibody. Kinoshita et al., (*J. Hepatol.*

2010. 53: 903-910) shows that a proportion of the CD68-positive Kupffer cell populations is negative to F4/80 antibody staining. Therefore 30 % Kupffer cell negative sporozoite crossing in Fig. 1C (Tavares et al., 2013) may not be accurate. Frevert identified morphologically macrophage-like cells as a Kupffer cells, which may include F4/80 negative Kupffer cells. We identified the Kupffer cell CD68, a marker of phagocytic macrophage, as a major receptor for the sporozoite liver invasion because CD68 KO mice has 71 % reduction of parasite liver invasion. Therefore, we described "Sporozoites leave the circulation mainly by traversing Kupffer cells..." and used supporting references such as Pradel et al. (2001): p1160, 'Parallel *in vivo* studies show that sporozoites traverse Kupffer cells, but not endothelia. Taken together, these results suggest that malaria sporozoites passage through Kupffer cells before hepatocyte invasion.'

4- I'm surprised by the fact that the authors are using CFA as adjuvant (Grumpstrup-Scott J, Greenhouse DD. 1988. NIH intramural recommendation for the research use of complete Freund's adjuvant. ILAR News 20:9). Was this adjuvant chosen because the fragments of PbGAPDH present low immunogenicity? The protocol of immunization/challenge should be repeated with an adjuvant that could be potentially used in humans (timeframe: ~1.5 mo)

We were not aware of this NIH recommendation about the CFA. As per this reviewer's suggestion, we repeated immunization and challenge assays for the epitope peptide using AddaVax™ (Invivogen), a squalene-based oil-in-water nano-emulsion with a formulation similar to MF59® that has been licensed in Europe for adjuvanted flu vaccines. (O'Hagan. 2007. *Expert Rev Vaccines*. 6: 699-710). We did two independent experiments and had similar results to those using CFA. The pooled 5 experiments are reported in Fig. 6D of the revised manuscript.

5- Control using sporozoites in the WB of figures 1A (a-P39) and 5B (a-fragments of GAPDH) are missing. This is important because antibodies against P39 recognize another band (70-100 kD) in sporozoites (Fig. 1A; Cha et al, JEM 2016))

The 70-100 kDa band in Fig. 1A of Cha et al, JEM 2016 is a background produced by anti-sera against the recombinant P39 bacteriophage (not against the peptide). The antibody against the synthetic P39 peptide does not bind to this band size. See the Fig. 1A of the revised manuscript.

In addition, the loop sequence between the two cysteines of P39 matches almost perfectly a fragment of PBANKA_1317200 and only partially the region of peptide 41 (GAPDH, Fig. 4A)

> PBANKA_1317200.1-p1 | transcript=PBANKA_1317200.1 | gene=PBANKA_1317200 | organism=Plasmodium_berghei_ANKA | gene_product=NAD(P) transhydrogenase, putative | transcript_product=NAD(P) transhydrogenase, putative | location=PbANKA_13_v3:722581-726493(-) | protein_length=1201 | sequence_SO=chromosome | SO=protein_coding | is_pseudo=false Length=1201 Score = 20.8 bits (42), Expect = 6.0, Method: Composition-

based stats. Identities = 7/8 (88%), Positives = 8/8 (100%), Gaps = 0/8 (0%)

```
Query 1 CAIVYAYDPC 8
      +IVYAYDP
Sbjct 863 SIVYAYDP 870
```

```
P39 1 -----CAIVYAYDPC----- 10
```

```
||::||:
```

```
PbGAPDH 301 KAGLALNDNFFKIVSWYDNEWGYSNRLLDLAIHITKN 337
```

We agree that there is significant sequence identity between P39 and PBANKA_1317200. This protein has 1201 amino acids that has a calculated molecular weight of 135.2 kDa. However as shown in the Western blot of Fig. 1A of the revised manuscript, the anti-P39 antibody does not recognize a protein of this size. Instead, the anti-P39 antibody recognizes a ~40 kDa band which was identified as PbGAPDH (Cha et al. 2016). Western blotting in Fig. 1B of Cha et al. 2016 shows 25 – 70 kDa range only, therefore additional experimental data showing full range of the blot added in this manuscript.

6- It would be important also to perform the experiments of figure 2 using antibodies against P61 and P52. Albeit less protective than P39, they could target the N-terminal domain of GAPDH, which is much less conserved than the C-terminal domain when comparing the *P.falciparum* and human sequences, or *P.berghei* and rodent sequences. Timeframe (2 weeks).

This is a good suggestion. Please see Fig. 3B of the revised manuscript.

7- Pre-patent period should be included in the figures 1D, 5C and table I to assess the level of control infection.

As suggested by this reviewer, pre-patent periods are listed in Figs. 2B and 6D of the revised manuscript but not for Table 1 because too few immunized mice were infected compared to the control immunization group.

August 31, 2018

RE: Life Science Alliance Manuscript #LSA-2018-00111-TR

Dr. Marcelo Jacobs-Lorena
Johns Hopkins Bloomberg School of Public Health
Molecular Microbiology & Immunology
615 N. Wolfe Street, E5132
Baltimore, MD 21205

Dear Dr. Jacobs-Lorena,

Thank you for submitting your revised manuscript entitled "IDENTIFICATION OF PLASMODIUM EPITOPES FOR THE INHIBITION OF SPOROZOITE LIVER INVASION". We would be happy to publish your paper in Life Science Alliance pending final revisions necessary to meet our formatting guidelines and address the remaining concerns from the referees (text changes only).

While refs #1 and #2 are overall satisfied with the revision, ref #3 points to a number of specific cases where the text still needs to be rephrased in order to properly acknowledge the existing literature and the nature of the assays used. In addition, the referee maintains that the current title overstates the findings when referring to liver invasion and consequently asks that you change it. Given the specific concerns about the type of experiments presented in the study I agree with the referee and would ask that you to rephrase the title to reflect this.

Please also address the following points:

- > The red arrows in Fig1B, Fig3B, Fig6B are not mentioned in the legends, please specify what they are.
- > Please include scale bars in Fig3C and Fig4B
- > Please provide source data for Fig4F (and ideally for the rest of the blots as well if you have the data readily available)

A. FINAL FILES:

-- High-resolution figure, supplementary figure and video files uploaded as individual files: See our detailed guidelines for preparing your production-ready images, <http://life-science-alliance.org/authorguide>

B. MANUSCRIPT ORGANIZATION AND FORMATTING:

Full guidelines are available on our Instructions for Authors page, <http://life-science-alliance.org/authorguide>

Sincerely,

Andrea Leibfried PhD
Executive Editor

Life Science Alliance

Reviewer #1 (Comments to the Authors (Required)):

In this revised version of their manuscript, the authors addressed most of my previous criticisms.

I only have a few comments left:

- 1) the title appears to be different in the ms text and the website. Please check carefully.
- 2) The sentence "In conclusion, data from Table 1 suggest that the efficacy of the antibodies against G6-1-20 and G6-41-60 are as protective as the anti-P39 antibody..." should be revised.
- 3) I would suggest to show the AddaVax data as a Fig 6E instead of pooling all the data in Fig6D.

Reviewer #2 (Comments to the Authors (Required)):

The authors improved the manuscript, mainly by toning down their statements. Some minor issues should still be addressed:

I would suggest the authors pass the manuscript by a native speaker to work over the introduction, which reads like a patchwork of sentences.

Title misses a word between first and second word

Abstract:

There is no such thing as a 'malaria sporozoite', there are Plasmodium sporozoites or Sporozoites of malaria causing parasites or Liver infecting Plasmodium sporozoites. Please rephrase.

Delete 'protective' from last sentence.

The reference to 'Sturm et al., 2006' in intro is still wrong

In the discussion please state 85% instead of 84,7% etc

Reviewer #3 (Comments to the Authors (Required)):

Reviewer #3 (Comments to the Authors (Required)):

The manuscript entitled « Identification of Plasmodium GAPDH vaccine antigens that target sporozoite liver invasion » by Cha and collaborators documents the identification of two 20-mer amino acid sequences from PbGAPDH that when conjugated to KLH and used with CFA/IFA to immunize Swiss mice (4 immunizing doses), passively and/or actively protected mice against sporozoite infection. In addition, they show the binding of region G6 of GAPDH to CD68, but not of the two protective fragments G6-1-20 and G6-41-60 (data not shown).

I have a few comments about interpretation of results and experimental data.

1- The title is misleading since no experiment showing inhibition of liver invasion (crossing of the sinusoidal barrier) is presented. In this manuscript, all readouts are related to blood infection (prevalence). In the previous manuscripts, the authors used "entry?" in macrophages as a proxy of cellular transmigration. Anti-P39, P61 and P52 antibodies inhibit "entry" but increase attachment to macrophage (Fig. 2B, Cha et al, JEM 2016). On the other hand, P39, P61 and P52, inhibited both entry and attachment (Fig. 2AB, Cha et al , JEM 2015), indicating that the mechanism of protection could be other than liver invasion, eg, increased phagocytosis via Fc receptor. Moreover, what the authors call "entry" in their assay is a measure of intracellular sporozoites, i.e., the total of sporozoites that invade macrophages plus those that were phagocytosed. However, since macrophages can kill intracellular sporozoites the readout at 2h reflects intracellular survivors. Importantly, this is a non-physiological assay since at 2h no substantial numbers of sporozoites are found inside/ traversing Kupffer cells in vivo. Please perform an experiment showing that immunization using GAPDH inhibits sporozoite liver invasion (translocation to the hepatic parenchyma) or change the title.

#Mimotope peptides bind to the Kupffer cell CD68, and the anti-P39 antibody binds to the sporozoite surface GAPDH. Incubation of the mimotope peptide at 0.4 mg/ml concentration may activate CD68 scavenging machinery for peptide uptake, which may competitively inhibit sporozoite attachment and entry into Kupffer cells. On the other hand, antibody binding to sporozoite surface does not inhibit CD68 machinery, however antibody binding to the sporozoite surface GAPDH epitope may inhibit interaction with CD68 by directing to the Fc receptor, which may lead sporozoite not to initiate Kupffer cell entry. Fig. 2B of Cha et al., 2016 shows that antibody binding to sporozoite GAPDH did not activate phagocytosis via Fc receptor. If it did, sporozoite attachment in the anti-mimotope antibody-treated groups should have decreased as compared to the control antibody-treated group (Fig. 2B, right panel).

A- The cells used in the Fig. 2B are peritoneal macrophages, not Kupffer cells. According to Eddie Wisse, "Kupffer cells and peritoneal macrophages are different types of cells, Blood cells 4(1-2):319-24, 1978)". Please also check Movita et al, JLB 92, 2012, for differences between KCs and peritoneal macrophages.

#In any case, any decrease of Kupffer cell traversal will result in decrease of liver invasion.

B- This statement does not seem to be exact, since depletion of KCs by clodronate does not decrease liver invasion. In addition sporozoites can directly traverse endothelial cells (Tavares et al, JEM 2013).

#The main aim of this work was to identify protective GAPDH epitopes. Neither the title nor the experiments in this manuscript address the detailed mechanism of decreased infection.

C- This is understandable but why the title is stating that protective antibodies inhibit sporozoite liver invasion?

Since there is no experiment showing inhibition of sporozoite liver invasion by the protective antibodies in this manuscript, the title should be either reformulated or the authors should provide experimental evidence of inhibition of liver invasion (entry in the liver parenchyma) to justify the title, like previously suggested.

In addition, GAPDH as pointed by the authors could be expressed on the surface of other plasmodial stages ("In Plasmodium, surface-GAPDH has been reported in multiple stages"), including invasive merozoites. Therefore protective antibodies could eventually target them. The readout of protection, prevalence of blood infection after sporozoite challenge, does not

allow concluding that antibodies are specifically inhibiting liver invasion.

2- The introduction is categorical, however the cited references do not sustain the authors' statements. For example, the paper of Cerami et al, 1994 shows that CSP binds in vivo to the microvilli of hepatocytes (Please see also Cerami et al, Cell 1992) and not to stellate cells.
#Thank you for pointing this out. This was changed.

D- You are welcome.

3- The next sentence, "Sporozoites leave the circulation mainly by traversing Kupffer cells..." is referenced by 6 manuscripts. Out of this 6, only the paper of Frevert et al, Plos Biol 2005 quantified the dynamic process of sinusoidal extravasation, but without the spatial resolution or appropriate markers to determine whether sporozoites were traversing Kupffer cells (KC) during this translocation. More recently, Tavares et al, JEM 2013 using a method to detect KC traversal by dynamic imaging showed that at least around 25% of estimated extravasation events are related to direct traversal of KC. This paper is much more relevant than the last cited manuscript (Sturm et al, Science 2006), which refers to the intravasation of merozoites, not to the extravasation of sporozoites. Please correct these two sentences (stellate cells and KC traversal).

#We specified references in more detail in the text. Of note, Frevert et al. (2005) and Tavares et al. (2013) used transgenic mouse strains that express GFP in the endothelial cell lining. Tavares et al. used F4/80 antibody to detect Kupffer cell using intravital confocal microscopy. However not all Kupffer cells are detected by the F4/80 antibody. Kinoshita et al., (J. Hepatol. 2010. 53: 903-910) shows that a proportion of the CD68-positive Kupffer cell populations is negative to F4/80 antibody staining. Therefore 30 % Kupffer cell negative sporozoite crossing in Fig. 1C (Tavares et al., 2013) may not be accurate.

E- As far as I understand, Kinoshita et al do not characterize any F4/80 negative population of KCs in their paper. On the contrary, they used F4/80+ cells to characterize two subpopulations of KCs in the mouse liver. The F4/80 is the most commonly used marker of rodent KCs (reviewed in Mowat et al, Nat Med 2017). This statement can be verified in the classical paper of Loyd et al, JIM 2008:

"We conclude that parenchymal macrophages are the dominant immune cell population in healthy liver, far outnumbering T cells, and that these cells are almost exclusively all F4/80+ »

or yet, in the paper of Movita et al, JLB 2012:

« Further evaluation of the phenotype of murine Kupffer cells by flow cytometry showed one population of cells that was defined as CD45+□CD11c□-F4/80highCD11b low »

Therefore, if the authors are aware of a KC population that is F4/80 negative, please specify. On the other hand, CD68 is not a specific marker of rodent KCs (Mowat et al, Nat Med 2017), but is expressed by monocytes, dendritic cells, adipocytes and many other non-myeloid cells (Chistiakov et al, Lab Invest 2017).

Besides, this does not solve the problem that the phrase ""Sporozoites leave the circulation mainly by traversing Kupffer cells..." is based on the manuscript by Frevert et al, which analysed four crossing events of weakly fluorescent lys-gfp KCs, and seven events of crossing through "auto-fluorescent?" rat KCs using wide-field microscopy. Frevert wrote in 2006 (Cell Micro, 8 pg 1538) "wide-field microscopy does not provide the resolution necessary to determine the exact position of KCs in the sinusoid so that the ultimate proof of sporozoite passage through KCs directly into the space of Disse has been difficult to produce".

#Frevert identified morphologically macrophage-like cells as a Kupffer cells, which may

include F4/80 negative Kupffer cells.

F- As stated before, please specify the publication that characterizes this presumable F4/80-negative KC population. Otherwise this remark is only speculative and should not be presented as justification, principally to introduce wrong concepts in the introduction. Please remark that some F4/80+ KCs are not lys-gfp+ , but not the contrary with the exception of brightly fluorescent circulating cells (Tavares et al, 2013; Fig.1). In addition, punctual cell autofluorescence cannot be more informative than F4/80-membrane staining. Thus, in my opinion and of the many other researchers, F4/80 is still the best marker of KCs in the mouse liver.

Finally, excluding the papers that are not quantitative, the authors are comparing 4 events of crossing through lys-gfp+ KCs and 7 events through autofluorescent KCs? (Frevert et al, Plos Biol 2005; 100% of crossing through KCs using wide-field microscopy) versus the analysis of 60 crossing events using F4/80+ labeled KCs where 40% of translocation occurs without the presence of a F4/80+ KC at the site of crossing, and of 16 events, using a specific marker of Kupffer cell traversal, which allowed the estimation that ~25% of total translocations occurs via KC traversal (Tavares et al, JEM 2013, using spinning disk confocal microscopy). Therefore, I'm surprised that the authors omitted the most accurate quantification of KC traversal during sinusoidal barrier crossing from the references, insisting in the notion that "sporozoites leave the circulation MAINLY by traversing KCs".

Please correct accordingly, otherwise this statement is misleading.

#We identified the Kupffer cell CD68, a marker of phagocytic macrophage, as a major receptor for the sporozoite liver invasion because CD68 KO mice has 71 % reduction of parasite liver invasion.

G- Again the authors only cite data that is convenient for the logic of their story, but omit an important result from their own manuscript that allows another interpretation of the role of CD68 in sporozoite infection.

"In CD68-knockout mice invasion (should be infection, since they measured parasite liver load not crossing of sinusoidal barrier) of the liver was reduced by more than 70 % of sporozoites compared to wild type mice (Cha et al, 2015)". However, depletion of phagocytic cells, including KCs, by clodronate treatment could revert this decreased liver infection showing that CD68 is not necessary for liver invasion (Cha et al., 2015). Please add this important information in the manuscript.

#Therefore, we described "Sporozoites leave the circulation mainly by traversing Kupffer cells..." and used supporting references such as Pradel et al. (2001): p1160, 'Parallel in vivo studies show that sporozoites traverse Kupffer cells, but not endothelia. Taken together, these results suggest that malaria sporozoites passage through Kupffer cells before hepatocyte invasion.'

H- Crossing experiments of Pradel et al, (2001) are all performed in vitro, without blood circulation. The phrase "Parallel in vivo studies show that..." is not referenced, and the same group published this data only in 2005 analysing 4 and 7 crossing events as explained above. So, I don't think this reference support that "sporozoite leave the circulation mainly by traversing KCs..." more than the papers that quantified this phenomenon in vivo (Frevert and Tavares).

Please introduce precisely the published literature, otherwise the paper is misleading.

Please also avoid the shortcuts between experimental data and interpretation. Therefore throughout the manuscript do not use liver invasion (entry in the liver parenchyma), when the readout is liver or blood infection. For example:

« Not only do these peptides selectively bind to the Kupffer cell surface but importantly, also competitively inhibit sporozoite Kupffer cell entry and access to the liver parenchyma. »

There is no data substantiating that the peptides inhibit the access of sporozoites to the liver parenchyma, this is pure speculation.

September 7, 2018

RE: Life Science Alliance Manuscript #LSA-2018-00111-TRR

Dr. Marcelo Jacobs-Lorena
Johns Hopkins Bloomberg School of Public Health
Molecular Microbiology & Immunology
615 N. Wolfe Street, E5132
Baltimore, MD 21205

Dear Dr. Jacobs-Lorena,

Thank you for submitting your Research Article entitled "IDENTIFICATION OF PLASMODIUM GAPDH EPITOPES FOR GENERATING ANTIBODIES THAT INHIBIT MALARIA INFECTION". It is a pleasure to let you know that your manuscript is now accepted for publication in Life Science Alliance. Congratulations on this interesting work.

The final published version of your manuscript will be deposited by us to PubMed Central (PMC) as soon as we are allowed to do so, the application for PMC indexing has been filed. You may be eligible to also deposit your Life Science Alliance article in PMC or PMC Europe yourself, which will then allow others to find out about your work by Pubmed searches right away. Such author-initiated deposition is possible/mandated for work funded by eg NIH, HHMI, ERC, MRC, Cancer Research UK, Telethon, EMBL.

Please also see:

<https://www.ncbi.nlm.nih.gov/pmc/about/authorms/>

<https://europepmc.org/Help#howsubsmanu>

DISTRIBUTION OF MATERIALS:

Again, congratulations on a very nice paper. I hope you found the review process to be constructive and are pleased with how the manuscript was handled editorially. We look forward to future exciting submissions from your lab.

Sincerely,

Andrea Leibfried PhD
Executive Editor

Life Science Alliance